# The Metabolomic Approach for the Screening of Endometrial Cancer: Validation from a Large Cohort of Women Scheduled for Gynecological Surgery

**DOI:** 10.3390/biom12091229

**Published:** 2022-09-02

**Authors:** Jacopo Troisi, Antonio Mollo, Martina Lombardi, Giovanni Scala, Sean M. Richards, Steven J. K. Symes, Antonio Travaglino, Daniele Neola, Umberto de Laurentiis, Luigi Insabato, Attilio Di Spiezio Sardo, Antonio Raffone, Maurizio Guida

**Affiliations:** 1Department of Medicine, Surgery and Dentistry, “Scuola Medica Salernitana”, University of Salerno, 84081 Baronissi, Italy; 2Theoreo Srl, Via Degli Ulivi 3, 84090 Montecorvino Pugliano, Italy or; 3Department of Chemistry and Biology, “A. Zambelli”, University of Salerno, 84084 Fisciano, Italy; 4Hosmotic Srl, Via Raffaele Bosco 178, 80069 Vico Equense, Italy; 5Section on Maternal-Fetal Medicine, Department of Obstetrics and Gynecology, University of Tennessee College of Medicine, Chattanooga, TN 37403, USA; 6Department of Biology, Geology and Environmental Sciences, University of Tennessee College of Medicine, Chattanooga, TN 37403, USA; 7Department of Chemistry and Physics, University of Tennessee at Chattanooga, Chattanooga, TN 37403, USA; 8Anatomic Pathology Unit, Department of Advanced Biomedical Sciences, University of Naples Federico II, 80138 Naples, Italy; 9Gynecology and Obstetrics Unit, Department of Public Health, University of Naples Federico II, 80138 Naples, Italy; 10Division of Gynaecology and Human Reproduction Physiopathology, Department of Medical and Surgical Sciences (DIMEC), IRCCS Azienda Ospedaliero-Univeristaria di Bologna. S. Orsola Hospital, University of Bologna, Via Massarenti 13, 40138 Bologna, Italy; 11Gynecology and Obstetrics Unit, Department of Neuroscience, Reproductive Sciences and Dentistry, University of Naples Federico II, 80138 Naples, Italy

**Keywords:** endometrial cancer, metabolomics, oncological screening, ensemble machine learning

## Abstract

Endometrial cancer (EC) is the most common gynecological neoplasm in high-income countries. Five-year survival rates are related to stage at diagnosis, but currently, no validated screening tests are available in clinical practice. The metabolome offers an unprecedented overview of the molecules underlying EC. In this study, we aimed to validate a metabolomics signature as a screening test for EC on a large study population of symptomatic women. Serum samples collected from women scheduled for gynecological surgery (*n* = 691) were separated into training (*n* = 90), test (*n* = 38), and validation (*n* = 563) sets. The training set was used to train seven classification models. The best classification performance during the training phase was the PLS-DA model (96% accuracy). The subsequent screening test was based on an ensemble machine learning algorithm that summed all the voting results of the seven classification models, statistically weighted by each models’ classification accuracy and confidence. The efficiency and accuracy of these models were evaluated using serum samples taken from 871 women who underwent endometrial biopsies. The EC serum metabolomes were characterized by lower levels of serine, glutamic acid, phenylalanine, and glyceraldehyde 3-phosphate. Our results illustrate that the serum metabolome can be an inexpensive, non-invasive, and accurate EC screening test.

## 1. Introduction

In the past three decades, there has been a worldwide increase in the incidence of endometrial cancer (EC) [1,2,3], and it is currently the most common gynecological neoplasm in high-income countries [3,4,5]. As stated by the GLOBOCAN cancer statistics, >380,000 new cases and approximately 90,000 deaths were attributed to EC in 2018 alone [6]. By 2030, EC incidence is predicted to be 55% greater than cases recorded in 2010 [7]. This sharp rise is likely due to the greater prevalence of associated risk factors, including obesity, diabetes, and metabolic syndrome; longer life expectancies will also be associated with increased incidence of EC [8].

The most common clinical presentation of the disease is postmenopausal bleeding [9,10,11]; however, only 5–10% of patients experiencing this symptom suffer from EC. Therefore, to confirm the diagnostic suspicion, women undergo various tests, with most of them being expensive, invasive, and sometimes difficult to perform. For example, a transvaginal ultrasound scan (TVS) is commonly employed as an initial investigation of potential EC, using endometrial thickness as an indicator of risk and to establish whether additional, more invasive procedures are needed. If abnormalities are detected, TVS is generally followed by endometrial biopsy or outpatient hysteroscopy. Patients are often reluctant to undergo these processes due to discomfort, pain, and procedure-related complications, including bleeding, potential risk of infection, and possible (but rare) uterine perforation. Furthermore, up to 25% of endometrial biopsies do not provide sufficient tissue for diagnosis in asymptomatic patients [10].

Five-year survival rates are strictly related to stage at diagnosis. At stage I FIGO (International Federation of Gynecology and Obstetrics), survival rates are >90%, and decrease to 20–26% at FIGO stage IV [8]. In addition, early diagnosis increases the chances of managing the disease through conservative treatments, thereby providing great benefits for women of childbearing age who wish to preserve their fertility as well as for those for whom surgery may be hazardous due to comorbidities or advanced age [12,13]. However, presently, no screening tests have been validated for EC. Therefore, the development of novel strategies for early cancer detection represents one of the greatest challenges in the field.

In this context, metabolomics analysis may be of pivotal support. Metabolomics is an emerging “omics” technology, which relies on the qualitative and quantitative assessments of the metabolites contained in a biological sample, combining high-throughput analytical technologies with advanced data analysis methods [14]. Compared to other biological measurements, the metabolome offers an unprecedented molecular overview of the mechanisms underlying the phenotype, as it reflects both the upstream changes occurring in the genome, transcriptome, and proteome, and the effects of environmental factors [15]. Indeed, although other approaches have the potential EC for screening and personalized medicine [16,17,18], none have the comprehensive ability to address mechanisms of action in the way that metabolomics does.

Recently, this promising approach has gained increased recognition in oncological research to such an extent that cancer metabolomics is now considered by many as an individual discipline [19]. Indeed, transformed cells exhibit several metabolic perturbations that mirror their dysregulated demands for uncontrolled proliferation [20,21,22]. As a consequence, patients affected by various types of cancers show characteristic metabolomic profiles that may be used to identify novel diagnostic and prognostic biomarkers as well as to discover new potential pharmaceutical targets [23]. Indeed, promising results have been obtained for diagnosis in several cancers, including endometrial, pulmonary, ovarian, breast, pancreatic, renal, hepatic, and cerebral carcinomas [24,25]. Accordingly, a growing body of evidence suggests that the onset and the progression of EC is associated with severe metabolic impairment, making this cancer particularly relevant to metabolomic investigations [26,27].

In a previous study, we described a serum metabolomic signature able to discriminate between women suffering from EC and healthy controls and from patients affected by other endometrial diseases [28]. Subsequently, we evaluated the effectiveness of the derived ensemble machine-learning (EML) algorithm as a potential EC screening test in a large cohort of postmenopausal women (i.e., >1400 subjects) [29]. The system exhibited excellent diagnostic performance with specificity, sensitivity, and accuracy all >99%. However, the low prevalence of EC in the general population was a major limitation of the study; only 16 participants were diagnosed with EC. In addition, due to the low prevalence of comorbidities and other potential confounding factors in the enrolled cohort, it was not possible to build an algorithm to account for the effects of such conditions. Thus, there was a need for additional studies in order improve the diagnostic capability of the EML algorithm. To that end, the aim of the present study was to strengthen the diagnostic efficiency estimate of a metabolomics-based classification algorithm in detecting the presence or absence of EC. This was achieved by enrolling a study cohort more likely to include a larger number of EC cases as well as a wider range of different conditions.

## 2. Materials and Methods

### 2.1. Study Design and Patient Enrollment

The study followed an a priori-defined study protocol and was designed as a single-center, observational, prospective, cohort study. The STrengthening the Reporting of OBservational studies in Epidemiology (STROBE) guidelines and checklist [30] were followed for reporting the study.

We performed two separate enrollments. The first 691 blood samples collected from EC-affected women and healthy controls occurred between May 2012 and January 2022. The second cohort enrolled 871 patients scheduled for gynecological surgery at the Division of Gynecology and Obstetrics, Department of Neurosciences, Reproductive Sciences and Dentistry, University of Naples Federico II between March 2019 and September 2020. Blood samples for metabolomic analyses were preoperatively collected from enrolled patients; women who did not provide consent for blood sample collection were excluded.

The whole study was carried out in accordance with the Declaration of Helsinki [31] and received approval from the Institutional Review Board of the University of Naples Federico II (N. 246/19). All enrolled women provided written informed consent for the use of their biospecimens for research purposes, and all data were anonymized to prevent the identification of the subjects.

### 2.2. Metabolomics Analysis

Extraction, purification, and derivatization of serum samples were performed using the MetaboPrep GC kit (Theoreo srl, Montecorvino Pugliano, Italy) as reported in Troisi et al. [28,29,32,33,34,35,36]. This sample preparation kit results in untargeted metabolomic profiles. Briefly, 50 µL of each serum sample was placed in separate Eppendorf tubes containing an alcohol-based extraction solution with 2-isopropyl malic acid as the internal standard. Tubes were then vortexed at 1250 rpm for 30 min and the solution was centrifuged for 5 min at 1600 rpm at 4 °C. Supernatant (200 µL) was transferred to new Eppendorf tubes, incubated with the purification solution, and vortexed at 1250 rpm for 30 s and subsequently centrifuged at 1600 rpm for 5 min at 4 °C. The resulting supernatants (175 µL) were placed in a 2 mL glass vial and freeze-dried overnight. Derivatization was performed in two steps: first, 50 µL of methoxylamine hydrochloride in pyridine solution was added and vortexed at 1200 rpm for 90 min; then, 25 µL of a derivatizing solution containing N,O-Bis (trimethylsilyl)trifluoroacetamide (BSTFA) was added and vials were vortexed again at 1200 rpm for 90 min. The 75 µL of derivatized metabolome was placed in a GC vial with a low-volume insert to facilitate the auto-sampler injection. Prior to injection into GC-MS, vials were centrifuged at 16,000 rpm for 5 min at 4 °C.

Derivatized samples (2 µL) were injected into the GCMS-2010SE (Shimadzu Corp., Kyoto, Japan). Chromatographic separation was achieved with a 30 mm × 0.25 mm CP-Sil 8 CB fused silica capillary column with 1.00 µm film thickness from Agilent (Agilent, J&W, Santa Clara, CA, USA), using helium as a carrier gas. The oven temperature was initially set at 100 °C and maintained for 1 min. Then, it was raised to 320 °C by 6 °C/min, with an additional 2.33 min of hold time. The gas flow was set to achieve a constant linear speed of 39 cm/s, while the split flow was set to 1:5. The mass spectrometer was operated with electron impact ionization (70 eV) in full scan mode with a range of 35–600 m/z, a scan velocity of 3333 amu/sec, and a solvent delay of 5 min. Peak identification was achieved setting the linear index (Kovats index [37] based on C10-C40 alkane mixture) difference max tolerance to 50, while the minimum matching for the NIST library search was set to 85%. Samples were divided into batches with each comprising 25 samples and being individually monitored using 4 different controls: an instrument blank injection, an injection of a standard mix, an injection of a pooled sample solution, and a duplicated injection of a randomly selected sample in the batch. Hexane (2 µL) was used for the instrument blank, and the standard mixture consisted of a solution of 15 molecules (organic acids, sugars, amino acids, steroids, and fatty acids) that was derivatized in the same manner as the samples. The pooled sample contained 2 µL of 50 randomly selected derivatized samples, while the duplicate injection was performed using a random sample from the batch.

The batches were considered validated only if four conditions were achieved: the solvent blank did not generate any peaks; peak areas of all analytes in the standard mix (normalized to the internal standard peak area) remained within 10% of the expected value; the variance among the peak areas (normalized to the internal standard) of the 100 highest peaks of the repeated injection was less than 15% of the first injection; and the pooled sample was assigned in the same area as the other pooled samples, i.e., <5% of the total area of a model constructed using all the samples analyzed.

Only gas chromatography–mass spectrometry signals consistently found in at least 80% of the samples were considered. Very low-intensity metabolite peaks, resulting from low concentration and therefore poor mass spectral quality, were not investigated further. In addition, signals that resulted from the same metabolites (e.g., sugars resulting in multiple derivatization products) were considered as independent features.

### 2.3. Statistical Analysis

The Shapiro–Wilk test was used to analyze clinical data distribution. Because the continuous variables were normally distributed, Student’s *t*-test was applied to determine *p*-values, while the comparison of percentages was performed via the χ^2^-test. An α-value of 0.05 was considered statistically significant. For metabolomics data analysis, chromatographic data were organized in a table containing one sample in each row and one variable (metabolite) in each column (dataset). Prior to statistical analysis and model building, the raw chromatographic data for each sample were transformed in the following way: the peak area of each feature was normalized to that of the internal standard, the logarithms of these values were calculated, and the resulting values were then autoscaled (mean-centered and divided by standard deviation of that variable).

### 2.4. Machine Learning Models

After samples met the criteria established above, the samples from the first enrollment (*n* = 691) were separated into training (*n* = 90), test (*n* = 38) (70:30 ratio), and validation (*n* = 563) sets. Sample size evaluation based on Troisi et al. [29] highlighted the need for at least 120 total subjects for the training and testing phase. The training set was used to train several classification models, of which 7 resulted in an accuracy > 65% and AUC-ROC > 0.85 with no overfitting. The classification models Naïve Bayes (NB), Generalized Linear Model (GLM), Fast Large Margin (FLM), Deep Learning (DL), Decision Tree (DT), Random Forest (RF), and Partial Least Squares Discriminant Analysis (PLS-DA) were selected and used to build an ensemble machine learning (EML) model. The test set was used to optimize the classification performance of each selected model, while the validation set was used to develop the EML and evaluate the best EML score threshold to identify EC positive cases. A second, independent validation set (*n* = 871) was used to blindly determine the overall classification performance of the developed EML algorithm.

During model training, a hyperparameter optimization step was included based on a grid search scheme. These were selected to maximize the models’ accuracy, also minimizing the overfitting risk. Moreover, a feature selection step, based on a genetic algorithm, was also included in the data analysis pipeline before each classification model.

Before the genetic algorithm, features were screened according to 3 criteria: (a) correlation, (b) stability, and (c) missing data. Regarding the correlation, the features that too closely, or not at all, mirrored the assigned class were excluded. Regarding the stability, we excluded features where >80% of values were identical. Features containing missing data were excluded.

The results of the seven trained classification models were statistically “ensembled” to create an EML algorithm using a voting scheme. The EML model constitutes the screening test referred to herein. To generate this model, each individual classification model was queried using metabolomic data from the validation set (*n* = 563) to generate a prediction (EC or CTRL) for a given sample. These votes were then weighted using both the cross-validation accuracy and the confidence (i.e., distance from classification margin or class centroid) of that model. The seven individual weighted votes for a given sample were then multiplied by “1” if that model predicted EC and by “−1” in the case of CTRL prediction. Next, a final, overall EML score was calculated for each sample by summing all of these adjusted single scores. The purpose of this mechanism is to have an EML score = 0 for a hypothetical sample where prediction for and against EC diagnosis were equal both in terms of number and weight. In contrast, an EML score > 0 indicates an EC-“positive” patient according to our model, while an EML score < 0 indicates a non-EC (CTRL) patient [28,29,32,33,34,35,36].

For each model, the area under the receiver operating characteristic (AUC-ROC) curve, sensitivity, specificity, positive and negative predictive values, positive and negative likelihood ratios, and accuracy were calculated after applying each model to the validation cohort (*n* = 563). These parameters were also evaluated for the EML score after the best threshold was established according to DeLong et al. [38] using a confusion matrix to summarize the results obtained in terms of true and false positive/negative classifications. Statistical analyses were partially performed using specifically designed R language codes (ver. 4.2.1, R Foundation, Vienna, Austria) and partially using Rapid Miner Studio ver. 9.10.0 (RapidMiner GmbH, Boston, MA, USA).

## 3. Results

Metabolomic profiles derived from samples obtained from an initial enrollment of subjects were used to train, test, and validate the effectiveness of an EML model in recognizing the presence of EC. This first validation, performed on a cohort of 563 individuals, included data from an unblinded experiment as, for all subjects, the EC status was known. Therefore, to perform an independent validation, a second blinded enrollment of 871 subjects, with unknown EC status but at risk of EC, was used to evaluate the efficiency, accuracy, and overall classification performance of the EML model. This second validation cohort consisted of subjects who had undergone gynecological surgery for various medical reasons at the Gynecology and Obstetrics Unit of the University of Naples “Federico II”.

For all 871 subjects in the second validation cohort, age, weight, height, smoke habits, metabolic pathologies (diabetes, hypertriglyceridemia, hypercholesterolemia, hyperuricemia, hypertension, vasculopathies), blood pressure, heart rate, and menometrorrhagia were recorded (Table 1). These parameters were normally distributed according to the Shapiro–Wilk test. These subjects were then divided into groups according to their pathological findings. A first group, defined as “non-cancer diseases”, included a total of 473 individuals who were further classified into 15 classes: healthy controls (CTRL, *n* = 171), endometriosis (E, *n* = 42), benign endometrial hyperplasia (EH, *n* = 17), high-grade squamous intraepithelial lesion (H-SIL, *n* = 28), low-grade squamous intraepithelial lesion (L-SIL, *n* = 12), myomas and/or polyps (M&P, *n* = 213), ovarian cyst (OC, *n* = 130), and uterine malformation (UM, *n* = 31). Another group, defined as “other cancers”, consisted of 101 patients suffering from malignancies other than endometrial cancer, including breast cancer (BC, *n* = 17), cervical cancer (CC, *n* = 19), ovarian cancer (OK, *n* = 44), vaginal cancer (VC, *n* = 13), and uterine sarcoma (US, *n* = 8). In the last group, 126 subjects diagnosed with endometrial cancer (EC) were included.

Ages were significantly different between the investigated classes. Indeed, EC patients were older, while “non-cancer disease” patients were younger (*p* < 0.001). Moreover, both weight and BMI were higher in EC patients, as well as the prevalence of hypertension, diabetes, hypertriglyceridemia, vasculopathies, and abnormal uterine bleeding. Endometrial thickness and heart rate were significantly higher in EC subjects compared to all the other classes.

Chromatographic separation of derivatized serum samples allowed the detection of up to 273 peaks in each specimen. Peaks that were not present in at least 80% of samples or with insufficient signals to be confirmed as metabolites by means of library comparison were excluded. This resulted in a total of 251 signals consistently detected.

Overall, 33,219 models based on 9593 feature subsets were built and tested to find the best combination of hyperparameters and metabolites able to maximize the classification accuracy of the investigated models. The best classification performance, among the individual models, during the training phase was recorded using the PLS-DA model (96% accuracy), whereas the worst was by FLM which, while having a very high sensitivity (100%), showed poor specificity (24%), resulting in 65% accuracy (Table 2).

The PLS-DA model was also used to graphically represent the sample separation achieved during the training phase (Figure 1). This resulted in no sample overlap, as further represented by the adequate cross-validation results (R^2^ = 0.97, Q^2^ = 0.71) and the strong statistical significance of the permutation test (*p* < 0.0055). In addition, the PLS-DA model was trained to calculate the statistical importance that each metabolite contributes to the overall class separation. These scores are called variable importance in projection (VIP) scores. Eleven metabolites reported a VIP score higher than 2.0, namely, urea, glycine, phenyl pyruvic acid, 3-hydroxybutyric acid, valine, serine, stearic acid, phenylalanine, proline, acetic, and glutamic acid (Figure 1B).

The relative abundance of metabolites in the two studied classes (CTRL and EC) was also assessed using the volcano plot (Figure 1E). Twelve metabolites showed both a fold change (FC) higher than 2 or lower than 0.5 and a *p*-value < 0.05. Three (glycerol, 3-hydroxybutyric and stearic acid) were higher in EC patients, while nine (glycine, phenyl pyruvic acid, serine, valine, urea, oxyproline, phenylalanine, glyceraldehyde 3 phosphate and gluconic acid) were lower in EC patients.

In Figure 2, the EML score distribution among the CTRL and EC samples of the validation cohort from the first enrollment is reported. The area under the ROC curve was 0.974 (95% CI: 0.884–0.999) when using the Youden index (sensitivity + specificity−1)-based best threshold value of 210.

Samples from the validation cohort taken from the second enrollment underwent a class assignment in terms of EC or CTRL by each trained classification model. After score calculation, the EML score was estimated summing the single scores. Figure 3 reports the score distribution among all the patients’ classes.

Table 3 reports the EML score mean, max, and min values in the studied patient classes. The table also reports the classification errors for each class. Almost all the patient groups showed a significantly lower (*p* < 0.05) EML score compared to the mean EML score of the EC group. Three exceptions were reported: uterine sarcomas (*p* = 0.13), vaginal cancer (*p* = 0.20), and L-SIL (*p* = 0.23).

## 4. Discussion

We aimed to validate a previously reported metabolomics signature [28,29] as a screening test for EC on a large study population of symptomatic women who underwent gynecological surgery for benign and malignant conditions. The screening test was based on an ensemble machine learning algorithm that sums all the voting results of seven different classification models, statistically weighted by each models’ classification accuracy and confidence. The EML-based screening test showed an error rate of less than 5% in identifying EC. Patients affected by other cancers were also identified at low error rates (0% for BC, for example). Three patient classes did not show a significantly different EML score compared to the EC patients (US, VC, and L-SIL). These three are the least represented classes which could help to explain this result, but these are also the classes with a greater histological overlap with EC and this should also be considered when interpreting this finding.

The age difference observed in the enrolled population does not seem to have played a determinant role in the direction of the training of the several machine learning algorithms. Indeed, the EML score resulted from subjects over and under 50 years of age was not statistically different in any of the analyzed classes (Appendix A). The metabolomics approach for the EC screening was already investigated in several studies by us [28,29] and other research groups [25,27,39]. Despite great efforts in EC screening research and the reported evidence of the feasibility of the metabolomics approach, heretofore, no clinical screening tool was available. Histological examination of endometrial specimens by hysteroscopy or dilation and curettage remains the gold standard in the diagnosis of EC. However, such a diagnosis may be limited as a screening system because of sampling invasiveness, sampling error (with particular regard to the early stage of the disease), high cost, procedure-related complications, and poor reproducibility of the histological examination even when performed by expert pathologists [40,41]. Moreover, some women are discouraged from undergoing this examination because they consider the procedure intimidating, a contributing factor for a late stage diagnosis [42]. This is unfortunate because there is a high survival rate when diagnosed at an early disease stage, highlighting the importance of a non-invasive, less intimidating screening system [6].

Metabolomics is a non-invasive, inexpensive, patient-friendly, and high-throughput technology that can simultaneously measure hundreds of different metabolites in a small volume of fluid or tissue [43]. Metabolomic impairments of cancer cells represent a well-known cancer hallmark [44]. These impairments are widely regarded as a robust basis of cancer metabolic profiles [35,36,45].

The results presented herein indicate that several serum metabolites are associated with EC. Specifically, the EC metabolome was characterized by lower levels of serine, glutamic acid phenylalanine, and glyceraldehyde 3-phosphate. These reductions align with the theories of Nobel laureate Otto Warburg, who speculated about the advantages of anaerobic glycolysis for cancer due to the higher speed of this metabolism and the low oxygen availability during rapid cancer growth (i.e., the Warburg effect) [46]. Indeed, metabolic impairment in cancer cells has been widely reported in recent years. For example, relatively recent evidence indicates that bypassing the tricarboxylic acid cycle (TCA) is an efficient strategy for cancer cells which effectively mitigates ROS production and reduces cancer cell apoptosis [20,45].

The reduction in different glycolytic enzymes has been reported in tumors, including EC [25,28,39]. This results in unconstrained energy production for tumor cells. Due to the Warburg effect, glycolysis does not proceed via the TCA, while pyruvate is shunted through lactate production aided by increasing lactate dehydrogenase (LDH) expression. LDH is considered a poor prognostic marker in many cancers because it reflects both high cytolysis levels and the high shunting rate of pyruvate [47].

One of the most important limitations in clinical application of the previously reported metabolomics signatures is the lack of validation on large patient cohorts as well as validation in real clinical settings. Herein, we report the results obtained in a large cohort of symptomatic women that could be interpreted as a clinical validation of the screening test in real conditions. We found a very high accuracy which exceeds that expected for a screening test. Other solutions for the screening of EC were proposed [48,49]. More than 10 years ago, Jacobs et al. [48] reported that endometrial thickness measured via transvaginal ultrasound could predict both EC and atypical endometrial hyperplasia. Indeed, analyzing the results of 48,230 women enrolled in the United Kingdom Collaborative Trial of Ovarian Cancer Screening, a sensitivity of 80.5% (95% CI 72.7–86.8) and specificity of 86.2% (85.8–86.6) were reported using an endometrial thickness cut-off of 5.15 mm. On the contrary, Kinde et al. [49], using the Papanicolaou test, performed massive parallel sequencing to identify DNA mutations from liquid biopsy smear specimens. They reported 100% accuracy to identify EC cases. Unfortunately, this evaluation was based on a very limited sample size (*n* = 24 EC cases in training and 14 cases for validation).

The metabolomics solution we propose is minimally invasive because it avoids uterine tissue sampling and does not require uncomfortable visits. Therefore, use of the metabolome for early screening could reduce discomforts to only the definitive diagnosis—which must always be performed by means of a pathological evaluation of the endometrial tissue. Further studies may be necessary to confirm these findings on even larger cohorts. Nevertheless, our results illustrate an approach that results in a suitable differentiation of EC with other conditions. Ultimately, the metabolomic approached could be broadly adopted for routine screening to support women’s reproductive health.

## Figures and Tables

**Figure 1 biomolecules-12-01229-f001:**
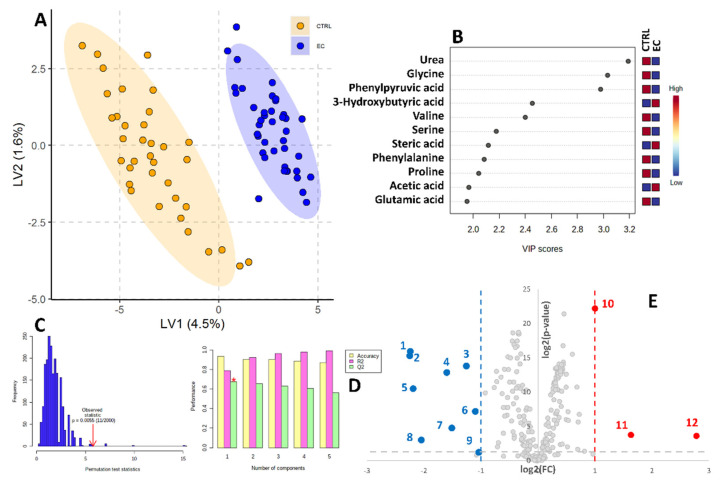
Partial Least Squares Discriminant Analysis (PLS-DA) based on serum metabolites determined by GC-MS. (**A**) Two-dimensional score plot showing clustering and separation between healthy CTRL serum profiles (orange) and endometrial cancer-affected patients’ profiles (blue) from the *n* = 90 training set. (**B**) Metabolites showing a VIP score > 2.0 in the PLS-DA analysis. (**C**) Permutation test results based on 2000 iterations. (**D**) PLS-DA classification performance using increasing number of latent variables. The red star indicates that the best model was achieved using only 1 variable. (**E**) Volcano plot reporting metabolite concentration fold changes and their statistical significance comparing CTRL vs. EC subjects among the second validation cohort. 1. Glycine, 2. phenyl pyruvic acid, 3. serine, 4. valine, 5. urea, 6. oxyproline, 7. phenylalanine, 8. glyceraldehyde 3 phosphate, 9. gluconic acid 10. glycerol, 11. 3-hydroxybutyric acid, 12. stearic acid.

**Figure 2 biomolecules-12-01229-f002:**
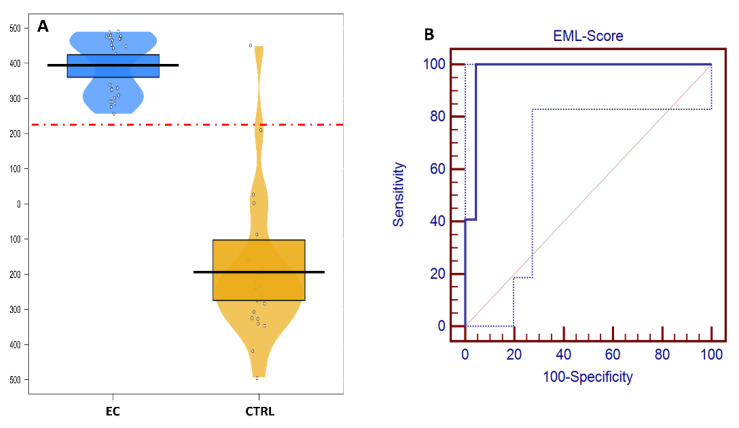
(**A**) Ensemble Machine Learning (EML) scores of healthy controls (CTRL, blue) and endometrial cancer (EC, orange)-affected patients; the red dashed line represents the Youden index-based optimized cut-off value. (**B**) Receiver operating characteristic (ROC) curve obtained by varying the cut-off value when applying the EML model to the test set. Dotted blue line represents the 95% confidence bounds.

**Figure 3 biomolecules-12-01229-f003:**
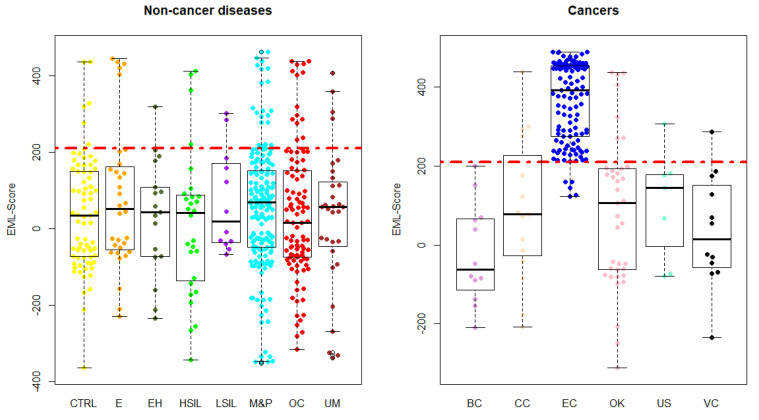
EML score distribution among the enrolled patients’ classes. Abbreviations used are CTRL: healthy controls; E: endometriosis; EH: endometrial hyperplasia; H-SIL: high-grade squamous intraepithelial lesion; L-SIL: low-grade squamous intraepithelial lesion; M&P: myomas and/or polyps; OC: ovarian cyst; UM: uterine malformation; BC: breast cancer; CC: cervical cancer; EC: endometrial cancer; OK: ovarian cancer; US: uterine sarcoma; VC: vaginal cancer.

**Table 1 biomolecules-12-01229-t001:** Characteristics of subjects enrolled in the second validation set (mean ± standard deviation or number and %). Abbreviations used are CTRL: healthy controls, EC: endometrial cancer, BMI: body mass index, bpm: beats per minute, SBP: systolic blood pressure, DBP: diastolic blood pressure.

	CTRL	Non-Cancer Diseases	Other Cancers	EC
Sample size (*n*)	171	473	101	126
Age (y)	45.4 ± 19.9	42.3 ± 12.2 * §	56.7 ± 12.7 * §	62.1 ± 10.3 *
Smoke (%)	23.5	31.5 * §	33.0 * §	17.7
Weight (Kg)	66.9 ± 12.3	67.2 ± 13.9 §	67.8 ± 15.0 §	79.2 ± 17.1 *
Height (cm)	160.5 ± 12.9	160.8 ± 20.7	153.3 ± 32.3	155.2 ± 31.1
BMI (kg/cm^2^)	27.7 ± 19.4	25.2 ± 5.1 * §	26.7 ± 6.2	30.5 ± 6.1
Hypertension (%)	22.8	14.6 §	39.6	53.2 *
Diabetes (%)	4.9	4.1 §	6.1 §	16.4 *
Hypercholesterolemia (%)	8.7	4.5 *	9.0	9.8
Hypertriglyceridemia (%)	0.0	0.6 §	2.0 *	1.6 *
Hyperuricemia (%)	0.0	0.6	2.0 *	0.0
Vasculopathies (%)	6.8	7.1 §	9.1 *§	13.1 *
Cholecystectomy (%)	6.8	3.8 *	4.0	6.5
Endometrial thickness (mm)	6.8 ± 5.0	8.0 ± 7.8 * §	5.0 ± 5.8 * §	13.0 ± 10.4 *
Abnormal uterine bleeding (%)	14.9	15.8 §	8.3 * §	37.7 *
SBP (mmHg)	118.9 ± 11.7	117.7 ± 9.8 §	120.3 ± 10.6	123.8 ± 13.4
DBP (mmHg)	76.6 ± 6.4	76.0 ± 7.0	77.7 ± 5.9	77.1 ± 7.1
Heart rate (bpm)	72.4 ± 6.5	73.2 ± 5.3 §	75.6 ± 7.3 §	77 ± 6.1 *

* Indicates *p*-value < 0.05 compared to CTRL; § indicates *p*-value < 0.05 compared to EC.

**Table 2 biomolecules-12-01229-t002:** Diagnostic performance of the individual and the ensemble machine learning algorithms used for classification among the validation set from the first enrollment. Abbreviations used are NB: Naïve Bayes, GLM: Generalized Linear Model, FLM: Fast Large Margin, DL: Deep Learning, DT: Decision Tree, RF: Random Forest, PLS-DA: Partial Least Square Discriminant Analysis, EML: Ensemble Machine Learning, S: sensitivity, Sp: specificity; PLR: positive likelihood ratio, NLR: negative likelihood ratio, NPV: negative predictive value, PPV: positive predictive value, A: accuracy, ND: not determinable.

Model	S	Sp	PLR	NLR	NPV	PPV	A
NB	0.74 ± 0.10	0.94 ± 0.06	12.53	0.28	0.76 ± 0.09	0.93 ± 0.06	0.83
GLM	0.90 ± 0.07	0.88 ± 0.08	7.20	0.11	0.88 ± 0.08	0.90 ± 0.07	0.89
FLM	1.00 ± 0.00	0.24 ± 0.10	1.31	0.00	1.00 ± 0.00	0.61 ± 0.09	0.65
DL	1.00 ± 0.00	0.63 ± 0.12	2.67	0.00	1.00 ± 0.00	0.77 ± 0.08	0.83
DT	0.95 ± 0.05	0.88 ± 0.08	8.05	0.06	0.94 ± 0.06	0.90 ± 0.07	0.92
RF	1.00 ± 0.00	0.29 ± 0.11	1.40	0.00	1.00 ± 0.00	0.61 ± 0.09	0.67
PLS-DA	0.93 ± 0.05	1.00 ± 0.00	ND	0.07	0.92 ± 0.05	1.00 ± 0.00	0.96
EML	1.00 ± 0.00	0.96 ± 0.04	23.00	0.00	1.00 ± 0.00	0.96 ± 0.04	0.98

**Table 3 biomolecules-12-01229-t003:** EML score means, min and max values among the various classes of patients enrolled in the second validation cohort. Abbreviations used are CTRL: healthy controls; CIN: cervical intraepithelial neoplasia; E: endometriosis; EH: endometrial hyperplasia; H-SIL: high-grade squamous intraepithelial lesion; L-SIL: low-grade squamous intraepithelial lesion; M&P: myomas and/or polyps; OC: ovarian cyst; UM: uterine malformation; BC: breast cancer; CC: cervical cancer; EC: endometrial cancer; OK: ovarian cancer; US: uterine sarcoma; VC: vaginal cancer. * Indicates statistically significantly different from (*p* < 0.05) EC subjects.

Class	N	Mean ± St Dev	Min	Max	Classification Errors
CTRL	171	40.6 ± 146.6 *	−363.4	435.6	5.6%
E	42	80.1 ± 187.0 *	−228.7	443.2	11.9%
EH	17	31.5 ± 152.0 *	−234.8	318.7	5.9%
H-SIL	28	13.0 ± 190.4 *	−342.3	411.2	14.3%
L-SIL	12	71.9 ± 133.6	−67.6	301.6	16.7%
M&P	213	47.4 ± 169.9 *	−351.6	461.8	9.9%
OC	130	37.5 ± 173.4 *	−314.6	437.0	10.8%
UM	31	29.0 ± 188.5 *	−336.9	406.0	12.9%
BC	17	−23.4 ± 126.7 *	−209.8	199.6	0.0%
CC	19	88.6 ± 184.2 *	−207.3	438.1	21.1%
OK	44	84.8 ± 192.0 *	−311.3	436.8	15.9%
US	8	103.0 ± 141.5	−79.4	306.0	12.5%
VC	13	35.2 ± 143.1	−234.7	286.8	7.7%
EC	126	363.0 ± 101.6	122.1	488.9	4.0%

## Data Availability

The data presented in this study are available on request from the corresponding author. The data are not publicly available due to ethical restrictions.

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
