# Peer review of "The Metabolomic Approach for the Screening of Endometrial Cancer: Validation from a Large Cohort of Women Scheduled for Gynecological Surgery"

_biomolecules, 2022, doi:10.3390/biom12091229_

Round 1

Reviewer 1 Report

I read with great interest the Manuscript titled “The metabolomic approach for the screening of endometrial cancer: validation from a large cohort of women scheduled for gynecological surgery”, which falls within the aim of the Journal.
In my honest opinion, the topic is interesting enough to attract the readers’ attention. The methodology is accurate, and the data analysis supports conclusions. Nevertheless, authors should include in the discussion section a consideration about some others potential endometrial cancer biomarkers such us non coding RNA including the following recent paper in the references section: PMID: 34830129, PMID: 34442102, PMID: 33808791 . Manuscript should be further revised by a native English speaker to improve clarity and readability. I want to inform You that I make a plagiarism check routinely, and I can confirm that Yours is an original writing.

Reviewer 2 Report

This study validated a previously reported metabolomics signature as a screening test for endometrial cancer on a large study population of symptomatic women who underwent gynecological surgery for benign and malignant conditions. Here are several concerns which would need to be addressed in the present manuscript:

 1.    Basically, screening test cannot replace diagnostic(confirmatory) test. Therefore, the serum metabolome cannot replace ‘histological examination of endometrial specimens by hysteroscopy or dilation and curettage’ although it can be an inexpensive, non-invasive, and accurate EC screening test. Endometrial biopsy with or without hysteroscopy is not screening test but diagnostic test. This concept must be corrected in the entire manuscript.

2.    Please explain in detail about ‘previously reported metabolomics signature’.

3.    Authors separated ‘serum samples collected from women scheduled for gynecological surgery (n=691)’ into training (n=90), test (n=38) and validation (n=563) sets. What are reasons of big difference among sample size of groups?

4.    The training set was used to train seven classification models. What is the reason that seven classification models are used? Please provide the evidences.

5.    Contents of Table 1 are different with the contents of text.

6.    In Figure 1, letters ‘D’ and ‘E’ are located incorrectly.

7.    Please explain ‘Youden index’.

8.    In 8 of 13 page, meaning of “Samples from the validation cohort…………..” is vague. Which cohort does validation cohort mean? 1st or 2nd ?

9.    In 9 of 13 page, please provide significance of Figure 3 and Table 3.

10.  It is difficult to understand the results in several points. Please explain in more detail the results.

11.  Please discuss the results of this study with the methods reported as screening tools of endometrial cancer.

Reviewer 3 Report

The authors have performed a study using metabolomics to validate the screening of endometrial cancer in a large cohort of women scheduled for gynaecological surgery.

The methods regarding metabolomic techniques and the applied statistics seem satisfactory as far as I am able to evaluate this.

What I would argue is the selection of cohorts and the rationale for  this being a validation of “screening” of endometrial cancer.

It is stated that this was an observational, prospective cohort study using a cohort of women enrolled from May 2012. However the regional Ethics confirmation (Institutional Review Board of the University of Naples Federico II (N. 246/19) ) imply that this was granted in 2019. Are there another approval from before 2012?

In the introduction and materials/methods there is a mixture of terminology: screening usually imply testing a large cohort of asymptomatic individuals to detect disease. The dominant insidence of EC is > 60 years , the majority with postmenopausal bleeding as incident symptom. When this is investigated the majority of cancers will be detected at early stage.

The cohorts used here are a mixture of asymptomatic women schedlueld for other types of surgery (including colposcopy and biopsy (?), hopefully not conization/hysterectomy, for LSIL and hysteroscopic procedures for uterine malformations) dominantly beeng <<60 years. Is this a cohort valid for EC screening? The risk of EC in these young patients is very low. 

Comparing metabolomics between postmenopausal patients with symptoms and Premenopausal women with other gynecological conditions does not seem meaningful in adressing a «screening» procedure. I would recommend comparing women with postmenopausal bleeding with and without cancer or postmenopausal women with symptoms versus without symptoms.

The introduction and the discussion should be rephrased regarding indications/conclusion.

Round 2

Reviewer 2 Report

1.    Authors mentioned in abstract as follows: “Patients are reluctant to EC screening tests because they are currently expensive, invasive, and painful.” However, “expensive, invasive, and painful” is characteristics of diagnostic(confirmatory) test such as histological examination of endometrial specimens by hysteroscopy or dilation and curettage’. Therefore, “screening tests” should be replaced with diagnostic tests.

2. Please discuss the results of this study with the methods reported as screening tools of endometrial cancer. Reply: As we reported in several sections, currently no screening tool is available for EC.

=> Authors mentioned in discussion as “Our results seem to be better than those reported for the most promising EC screening tools to date [48,49].” Please discuss the results of this study with the above methods reported as screening tools of endometrial cancer.

3. It is still difficult to understand the results in several points.

Reviewer 3 Report

The authors have not addressed my primary concern: they claim to “validate screening for endometrial cancer in a large cohort of symptomatic women”

This is not av screening validation because the population that is screened does not have  an indication of screening for EC, they are coming to hospital for totally different reasons, being at an age with so little risk of EC that screening is not feasible.

In the reply the author indeed states this as “As we reported in PMID: 32986110 we proposed our metabolomics signature of EC as a screening testing it on a large (>1400) asymptomatic women. In such a setting we were able to identify only 16 EC affected women.” So haphazard screening is not warranted!

They continue ”A larger study on asymptomatic women will be very expensive so we moved on a different setting, testing our signature on a cohort with a higher number of EC cases. This is the reason why we used the described cohort. Of course, we recognize that this is not a typical setting of a screening test application, but it can be considered a more stressful setting to prove the effectiveness of such a screening test.” –No, it is NOT. Finding that a metabolomics signature identified in women with diagnosed EC is distinct different from younger women with non-cancer gynecological diseases or even other gynecological malignancies does not validate this as a screening test for EC. This MUST be explained in the conclusion/abstract.

“Because, almost 90% women with EC reported postmenopausal vaginal bleeding. Nevertheless, there are several causes of vaginal bleeding, indeed, according to recent evidence only 9% of women reporting vaginal bleeding in post-menopause had EC. So, this sign is not a good predictor of EC. The reported metabolomics signature offers a better performance as a screening solution.” This cannot be stated from this present study. When 9% (mostly cited as 1/10 with postmenopausal bleeding have cancer this is in fact considered a rather STRONG risk predictor of cancer! The proposed score should be tested in a cohort of women with postmenopausal bleeding to see if it outperforms this.

This study CANNOT conclude that this metabolomics model is validated as a screening test of EC. It is merely different in different cohorts of women.
